# Detecting Benchmark Contamination Through Watermarking

## Abstract

Benchmark contamination undermines LLM evaluations, and existing post-hoc detection methods are heuristics and thus lack verifiable guarantees. We propose a proactive solution: embedding cryptographic watermarks into benchmarks *before* their release through question reformulation with a language model, and introduce a detection algorithm that overcomes tokenizer mismatches by aligning text prefixes to reliably identify the watermark signal in the suspect model. To validate our method, we pre-train 1B-parameter models on 10B tokens with controlled contamination of MMLU and ARC. The watermarking process preserves benchmark utility, while our test detects contamination with high confidence, achieving, *e.g.*, a $p$-value $< 10^{-5}$ for a mere 5% performance gain on 5000 MMLU questions.

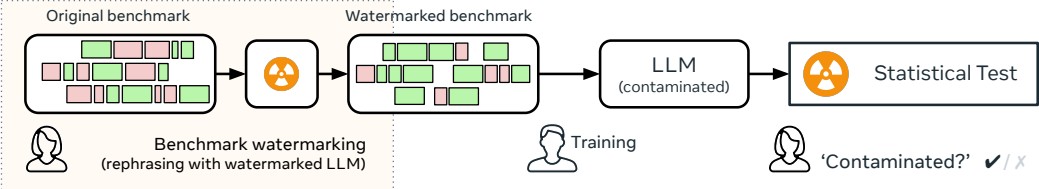

Figure 1: Problem overview. *Alice* is a benchmark provider. Before release, she rephrases the original benchmark dataset while embedding a watermark. *Bob* decides to train a model. The benchmark may contaminate Bob's model during training. Alice can give statistical evidence if her benchmark was used in training.

## 1 Introduction

In recent years, Large Language Models (LLMs) have demonstrated remarkable advancements in their capabilities (Brown et al., 2020; Touvron et al., 2023a). This advancement places increasingly greater emphasis on proper evaluation to both inform the state of LLM research and to guide future developments. To this end, a multitude of benchmark datasets such as (MMLU) (Hendrycks et al., 2020), School Math 8K (GSM8K) (Cobbe et al., 2021), and the AI2 Reasoning Challenge (ARC) (Clark et al., 2018), or more recently GPQA (Rein et al., 2023) and FrontierMath (Glazer et al., 2024), are developed to measure the model's capabilities in terms of general or specific knowledge, understanding, and scientific reasoning.

The reliability of LLM evaluation is critically undermined by benchmark contamination. While drops in performance on rephrased benchmarks such as the recent GSM8K variant (Zhang et al., 2024a) strongly suggest contamination, they do not provide definitive proof, leaving the community to debate the validity of a model's claimed capabilities. This uncertainty stems from the fundamental limitations of existing post-hoc detection methods. These methods analyze a model after training and are ultimately heuristics, *i.e.*, they rely on indirect evidence and observable patterns rather than direct proof of contamination. For instance, methods like Membership Inference Attacks (MIAs) rely on a held-out set from the same data distribution. This requirement is a paradox for public benchmarks: if such a set existed, it could simply serve as the new, uncontaminated benchmark. This gap reveals the need for a shift from post-hoc suspicion to proactive, verifiable proof. Instead of trying to find evidence of contamination after the fact, we argue that the community must embed a provable but undetectable signal into benchmarks before their release. Our work introduces such a framework.

We propose a novel strategy of embedding non-intrusive watermarks in the benchmark dataset before release. Our approach is inspired by Sander et al. (2024), who demonstrated that slightly distilling a watermarked LLM can be reliably detected, as the model retains identifiable traces of the watermark. We extend this idea to dataset watermarking, and when possibly different tokenizers are used by the watermarking and the suspect models. Our approach enables reporting both model performance on the benchmark and a reliable $p$-value as a contamination score, which relates to the False Positive Rate of the contamination test (see Proposition 1). If the reported $p$-value is low, the LLM's training data is likely contaminated with the benchmark dataset and the performance numbers should not be trusted as genuine. Our method requires only access to an LLM capable of rephrasing benchmark questions; see Figure 1 for an overview, and operates in a white-box setting to detect contamination, which would work for open source models and self auditing for closed ones. Our contributions are:

- **A proactive framework for detecting pre-training contamination.** We adapt the concept of watermark radioactivity (Sander et al., 2024), previously applied to instruction-tuning data, to the distinct and more challenging problem of detecting pre-training data contamination. Our method proactively embeds a secret statistical signal into benchmarks via a rephrasing LLM prior to their release – while safeguarding utility – in order to later provide provable evidence of contamination.

- **A robust, cross-tokenizer detection algorithm.** Recognizing the diversity of tokenizers in the LLM ecosystem, we introduce a novel detection algorithm that reliably identifies the watermark signal even when a suspect model uses a different tokenizer from the one used for embedding. This contribution significantly enhances the practical applicability of our method for auditing a wide range of models (Algorithm 1 in Sec. 3.2).

- **Extensive empirical validation and comparison.** We provide a large-scale empirical validation of this proactive detection method by pre-training models of up to 1B parameters on 10B tokens. Our experiments demonstrate strong correlation between the detection confidence and the performance inflation caused by contamination. They also show that our method is significantly more sensitive than comparable baselines like canaries. For instance, we detect contamination with a $p$-value below $10^{-5}$ when accuracy is inflated by only $5\%$ on MMLU, while correctly yielding $p$-values near 0.5 for uncontaminated models (Figure 3b and Table 1).

Our code will be made available to enable post-hoc text watermarking and contamination detection.

## 2 RELATED WORK

### 2.1 BENCHMARK CONTAMINATION DETECTION

Benchmark contamination is a significant concern that can lead to unreliable LLM evaluations (Singh et al., 2024; Balloccu et al., 2024). The issue is pervasive, as even rigorous decontamination efforts are not foolproof (Brown et al., 2020; Singh et al., 2024), and small amounts of contamination can significantly inflate performance (Jiang et al., 2024). The existence of this problem has been convincingly demonstrated by studies like the one by Zhang et al. (2024a). They crafted new questions for GSM8K and observed a significant drop in performance for most models, suggesting memorization of the original test set.

While a variety of post-hoc methods exist to detect contamination – from membership inference attacks (MIAs) (Carlini et al., 2022) to analyzing performance on reformulated questions (Yang et al., 2023; Duarte et al., 2024) or on reorederd answers (Oren et al., 2023)– they are fundamentally heuristics and face practical limitations. For instance, many require a held-out set, which is a paradoxical requirement for public benchmarks. Recent work has also shown that MIAs suffer from distribution shifts, further complicating efforts to reliably detect contamination (Meeus et al., 2025).

For active methods, the work most similar to ours is the contemporary paper by Rastogi et al. (2025), which approaches dataset membership inference by generating multiple rephrased versions of a benchmark. Each version is embedded with a unique watermark, yet only one is released publicly. By employing a paired t-test to compare the model's perplexity on the public version against the withheld private versions, their method identifies training inclusion based on the model's statistical preference for the specific public watermark. However, similar to other MIAs such as Maini et al. (2024), contamination detection for STAMP is restricted to the entity holding the private benchmark versions. Moreover, because it necessitates open-weight (or at least grey-box) access to the model, it

thus makes it impractical for evaluating proprietary models. In contrast, our method relies solely on the official version, enabling any party to perform the contamination test. Our approach is also close to hiding canaries inside the benchmarks as done in Srivastava et al. (2022), to which we compare directly as it does not necessitate any held-out.

We provide a detailed comparison and discussion of the trade-offs between all methods in 4.3.

## 2.2 Decoding-based watermarking & Radioactivity

**Overview.** Recent watermarking techniques for large language models (LLMs) involve altering either the probability distribution (Kirchenbauer et al., 2023a) or the method used for sampling the subsequent token (Aaronson & Kirchner, 2023; Kuditipudi et al., 2023). Detection of these watermarks is influenced by the entropy of the generated text (Christ et al., 2023; Huang et al., 2023), so further investigations propose watermarking only sections with high entropy, especially in code (Lee et al., 2023), while other studies explore "semantic" watermarks that rely on the semantic representation of the entire preceding text (Liu et al., 2023; Liu & Bu, 2024; Fu et al., 2024).

**Greenlist/redlist watermark.** Our work focuses on the watermarking scheme proposed by Kirchenbauer et al. (2023b), which modifies the logit vector during token generation based on a context window of $k$ previous tokens and a private key s. Both are hashed to serve as the seed for a random number generator (RNG) to create a "greenlist" of $\gamma|\mathcal{V}|$ tokens, where $\mathcal{V}$ is the vocabulary of the tokenizer, and $\gamma \in [0, 1]$. Logits of green tokens are incremented by $\delta$ to increase their sampling probability. Detection involves repeating the greenlist computation for each token of a text, incrementing a score by 1 if the token is in the greenlist, and performing a statistical test on the cumulative score. Under the null hypothesis $\mathcal{H}_0$ "the text is not watermarked with that scheme", this score follows a binomial distribution (Fernandez et al., 2023). A simple binomial test thus provides a $p$-value.

**Radioactivity of LLM watermarks.** Sander et al. (2024) show that fine-tuning language models on LLM-generated watermarked question-answer pairs can be detected with high confidence, as the model retains traces of the watermark bias. The authors adapt the original watermark detection tests to detect watermark "radioactivity" – a term first coined in Sablayrolles et al. (2020) for image data – depending on the access to the suspect model and its training data. Similar observations had been made in other scenarios. For instance, Gu et al. (2023) demonstrate that LLM watermarks can be intentionally distilled, and Gloaguen et al. (2025) examines model watermarks durability post-training. Zhao et al. (2023) introduce a signal in generated text that can be learned by other LLMs trained on it, and Jovanović et al. (2024) investigate watermark radioactivity for RAG. detect traces inserted during training, but the use case is largely different. Other work deal with embedding traces in models to detect it a Elhassan et al. (2025) embeds a LoRA-based watermark into weights during fine-tuning, aiming to identify the model generator.

## 3 Method

**Scope: threat model and access requirements.** Our work focuses on detecting contamination in LLMs that do next token-prediction with transformer-based models. Our method principally targets unintentional benchmark contamination, the common result of indiscriminate web scraping. Dedicated adversarial attacks, such as paraphrasing to evade detection, are not considered in this work. Moreover, our detection test operates in a white-box setting, requiring logit access of the suspect model. This enables transparent auditing of open-source models and supports verifiable self-reporting of contamination $p$-values for closed-source developers, fostering greater trust in evaluations.

We first focus in Section 3.1 on the task of rephrasing the questions of a benchmark dataset while embedding a watermark using Kirchenbauer et al. (2023b). Then, in Section 3.2, we show how to detect if a language model was trained on the watermarked benchmark.

### 3.1 Inserting watermark through question rephrasing

We use an instruct language model, denoted as $LM_{\text{rephrase}}$, which is assumed to be capable of rephrasing each question in the benchmark test set such that the rephrased version is logically equivalent to the original. This is a pretty light assumption as the task of rephrasing is considerably

easier than answering the question (Deng et al., 2023). $LM_{\text{rephrase}}$ generates token per token and at each step, takes as input a context, which is the concatenation of the system prompt, rephrasing instruction, the question to rephrase and the reformulation generated so far. Everything is tokenized into a sequence $\left(x^{(1)}, \ldots, x^{(t-1)}\right) \in \mathcal{V}^{t-1}$, where $\mathcal{V}$ is the vocabulary of the tokenizer.

$LM_{\text{rephrase}}$ outputs a logits vector $\boldsymbol{\ell}^{(t)} \in \mathbb{R}^{|\mathcal{V}|}$. The watermark embedding modifies $\boldsymbol{\ell}^{(t)}$ based on a secret key $\mathsf{s}$ (one per benchmark) and the watermark window $\left(x^{(t-k)}, \ldots, x^{(t-1)}\right) \in \mathcal{V}^k$

Specifically, following the method of Kirchenbauer et al. (2023b) detailed in Sec. 2.2, a secret-key cryptographic function hashes $\mathsf{s}$ as well as the the watermark window, which serves as a seed for a random number generator used to create a pseudo-random "greenlist" of tokens, comprising $\gamma = 50\%$ of the entire vocabulary $\mathcal{V}$, for which the logits are incremented by a quantity $\delta$ to form $\tilde{\boldsymbol{\ell}}^{(t)}$, thereby increasing their probability of being sampled. The logits vector is then transformed into a probability distribution $\mathbf{p}^{(t)} = \operatorname{softmax}(\tilde{\boldsymbol{\ell}}^{(t)}) \in [0, 1]^{|\mathcal{V}|}$, and the generation proceeds by sampling the next token $x^{(t)}$ from this distribution using a sampling procedure such as top-k sampling (Fan et al., 2018) or nucleus sampling (Holtzman et al., 2019). The selected token is appended to the context, and the process repeats. An example for the watermark embedding process is depicted in Figure 2a, with a detailed version with different strength of watermarking in Figure 5 of Appendix A.

**Detectability/utility tradeoff.** There is a common tradeoff in watermarking between detection and utility. In our case *detection* is the ability to have statistical evidence that the benchmark was used during training. We show in subsection 3.2 that it can be measured through the $p$-value, which can directly be linked to the False Positive Rate of the detection test (see Prop. 1). A lower $p$-value thus indicates a stronger detection signal, making it more likely to identify unauthorized usage. On the other hand, the *utility* of the watermarked benchmark is its ability to rank models and assess their performance on specific tasks. To preserve utility, we therefore require that models perform similarly on both the original and watermarked versions of the benchmark, allowing for accurate evaluation and comparison of model performance. Specifically, the benchmark dataset exhibits a proportion $\rho > 0.5$ of green tokens after rephrasing, the greater the easier detectability. For utility, we check if pre-trained models perform similarly on the original and rephrased versions.

We envision a practical workflow where benchmark creators are the final arbiters of quality. They can tune the watermark strength ($\delta$) and rephrasing model, and use a human-in-the-loop process to validate or select from multiple rephrased candidates, ensuring the benchmark's integrity.

## 3.2 DETECTING RADIOACTIVITY WITH A STATISTICAL TEST

To test a suspect model for contamination, we check for "radioactivity" by analyzing its predictions on the watermarked benchmark questions in a "reading mode" approach (Sander et al., 2024). That is, for each question, we forward the $N$ tokens that form the question and observe the $N$ predicted next tokens. The core idea is that a model contaminated with the benchmark data will have learned the statistical biases introduced by our watermark, causing its predictions to be skewed towards the watermark's "green list", as illustrated in Fig. 2b.

<table>
<tr>
<td>

**System prompt + instruction:**
"You are a problem rephrasing assistant [...]"

**Question:** "The rate of acceleration of an object is determined by the mass of the object and"

**Rephrased with watermark ($\delta = 4$):**
"What factor, aside from an object's mass, determines its acceleration?" (73% of green tokens)

</td>
<td>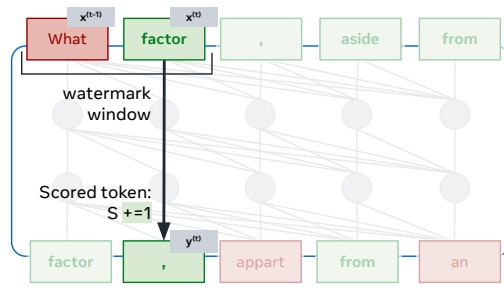</td>
</tr>
<tr>
<td align="center">(a) Embedding - benchmark rephrasing</td>
<td align="center">(b) Detection - statistical test</td>
</tr>
</table>

Figure 2: Method description. (Left) Watermarking benchmarks' questions using LLMs, as in subsection 3.1, with an example from ARC-easy. The quality of the question is maintained. (Right) Reading mode, as detailed in subsection 3.2. The upper sequence is the watermarked question, and the tokens bellow are top-1 predictions from the suspect model.

This allows us to formulate a powerful statistical test based on a clear null hypothesis, $\mathcal{H}_0$: an uncontaminated model's predictions are statistically independent of our secret watermark key, s. Since our scheme partitions the vocabulary into equally sized "green" and "red" lists for any context, a model under $\mathcal{H}_0$ should predict a green-list token with 50% probability. We can therefore count the number of green-list predictions, $S$, over a set of $\tilde{N}$ trials. To ensure each trial is independent and identically distributed (i.i.d.), we only score each unique context window once (Fernandez et al., 2023). This count follows a binomial distribution, $S \sim B(\tilde{N}, 1/2)$, and a significantly high score allows us to reject $\mathcal{H}_0$ and conclude the model is radioactive. The corresponding $p$-value is calculated using the regularized incomplete Beta function: $p\text{-value}(s) = I_{0.5}(s, \tilde{N} - s + 1)$.

A key practical challenge arises when the suspect model uses a different tokenizer ($T_2$) from the one used for watermarking ($T_1$), preventing direct comparison. Our method addresses this with an alignment procedure detailed in Algorithm 1. We tokenize the input with both $T_1$ and $T_2$ and only consider a prediction for scoring at "alignment points" where the text prefixes generated by both tokenizers are identical. At these points, we can safely regenerate the green list using the $T_1$ context. If the predicted token from $T_2$ also exists in $T_1$'s vocabulary, we score it (incrementing $S$ by 1 if it is green). This ensures our statistical test's validity across different tokenizers.

**Proposition 1.** *If we define "being contaminated" as having memorized the watermark (i.e., being radioactive), then the test $T_\alpha$ (that rejects $\mathcal{H}_0$ if the p-value is less than $\alpha$) correctly tests for contamination with a False Positive Rate (FPR) equal to $\alpha$.*

This proposition (proven in Appendix B) confirms that our $p$-value provides a theoretically grounded measure of contamination. While being radioactive is distinct from having an artificially inflated benchmark score, our results in Section 4 empirically demonstrate a strong correlation.

---

**Algorithm 1:** Reading Mode Scoring with Different Tokenizers

---

**Input:** Question $q$ from watermarked benchmark, Tokenizer $T_1$ for watermarking, Tokenizer $T_2$
    of suspect model $M$, tape $\mathcal{T}$ of already-scored watermark window, score $S$

Tokenize $s$ with $T_1$: $x_0, x_1, \ldots, x_{n-1}$;
Tokenize $s$ with $T_2$: $y_0, y_1, \ldots, y_{m-1}$;
Get top-1 predictions $\tilde{y}_1, \tilde{y}_2, \ldots, \tilde{y}_m$ from $M$;
**for** $i \leftarrow 0$ **to** $m - 1$ **do**
  **if** *there exists $j$ where text($y_0, \ldots, y_i$) = text($x_0, \ldots, x_j$)* **then**
   **if** $\tilde{y}_{i+1} \in T_1.vocab$ **and** $(x_{j-k+1}, \ldots, x_j) \notin \mathcal{T}$ **then**
    $S$ += Score($(x_{j-k+1}, \ldots, x_j, \tilde{y}_{i+1})$);
    $\mathcal{T}$.add($(x_{j-k+1}, \ldots, x_j)$);

---

## 4 RESULTS

### 4.1 BENCHMARK QUALITY AFTER WATERMARKING

**Set-up.** For the watermark embedding, we rephrase with Llama-3.1-8B-Instruct (Dubey et al., 2024) by default, with top-p sampling with p = 0.7 and temperature = 0.5 (default values on the Hugging Face hub), and the green/red watermarking scheme of Kirchenbauer et al. (2023b) with a watermark window $k = 2$ and a "greenlist" of size $\frac{1}{2}|\mathcal{V}|$ ($|\mathcal{V}|$ is the vocabulary size). We compare different values of $\delta$ when rephrasing: 0 (no watermarking), 1, 2, and 4. We choose to watermark ARC-Challenge, ARC-Easy, and MMLU due to their widespread use in model evaluation. In practice, one would need to watermark their own benchmark before release. For MMLU, we select a subset of 5000 questions, randomly chosen across all disciplines, to accelerate experimentation and maintain a comparable size to the other benchmarks. We refer to this subset as MMLU*. ARC-Easy contains 1172 questions, and ARC-Challenge contains 2372 questions. In Figure 5, we show the exact instructions given to the rephrasing model (identical for all benchmarks) and the results for different watermarking strengths. *We use a different watermarking key* s *for each benchmark.*

**Even strong watermarking retains benchmark utility.** We evaluate the performance of Llama-3.3-1B, Llama-3.3-3B and Llama-3.1-8B on the original benchmark and the rephrased version using

as similar evaluation as the one from the `lm-evaluation-harness` library (Gao et al., 2024). To check if the benchmark is still as meaningful, we check that evaluated models obtain a similar accuracy on the watermarked benchmarks and on the original version (see subsection 3.1). Figure 3a shows the performance on ARC-Easy. All models perform very similarly on all the rephrased versions of the benchmark, even when pushing the watermark to $80\%$ of green tokens. Importantly, they rank the same. Similar results are shown for MMLU* and ARC-Challenge in Figure 7 of Appendix A, although for MMLU*, we observe some discrepancies. For instance, when using a watermarking window size of 2 (subfig i), the performance of Llama-3.2-1B increases from $38\%$ to $42\%$ between the original and the other versions. However, we observe the same issue when rephrasing without watermarking in that case. As detailed in subsection 3.1, tuning the instruction specifically for each benchmark could help. Note that the choice of $\delta$ depends on the benchmark and the rephrasing model, and needs to be empirically tested. Performance of other models are given in app. D.

### 4.2 CONTAMINATION DETECTION THROUGH RADIOACTIVITY

We now propose an experimental design to control benchmark contamination, and evaluate both the impact on model performance and on contamination detection.

**Training set-up.** We train 1B standard decoder-only transformer architecture with the causal language modeling objective similar to the foundational T-decoder (Liu et al., 2018) and GPT (Radford & Narasimhan, 2018), adapted from the encoder-decoder architecture proposed in Vaswani et al. (2017) using `Meta Lingua` (Videau et al., 2024) and the code-base's default architectural choices. We train on 10B tokens from DCLM (Li et al., 2024), with the same tokenizer used to embed the watermark, unless stated otherwise (*e.g.*, in Sec. 4.5). The model architecture includes a hidden dimension of 2048, 25 layers, and 16 attention heads. The training process consists of 10,000 steps, using a batch size of 4 and a sequence length of 4096. Each training is distributed across 64 A-100 GPUs, and takes approximately three hours to finish. The optimization is performed with a learning rate of $3 \times 10^{-3}$, a weight decay of 0.033, and a warmup period of 5,000 steps. The learning rate is decayed to a minimum ratio of $10^{-6}$, and gradient clipping is applied with a threshold of 1.0.

**Contamination set-up.** Between training steps 2,500 and 7,500, we perform contamination by replacing a training batch from the DCLM corpus with a batch sampled from the shuffled concatenation of the three watermarked benchmarks. This batch replacement occurs periodically, specifically every 5000 / #ContaminationSteps training steps Each batch has batch size $\times$ sequence length $\times$ number of GPUs $= 4 \times 4096 \times 64 \approx 1\,\mathrm{M}$ tokens. As shown in Table 1, the concatenation of the three benchmarks is approximately 500k tokens, so each contamination is a gradient that encompasses all the benchmark's tokens. Any sample that contaminates the model is formatted as: `f"Question: {Question}\nAnswer: {Answer}"` In the experiments, #ContaminationSteps is refered as the

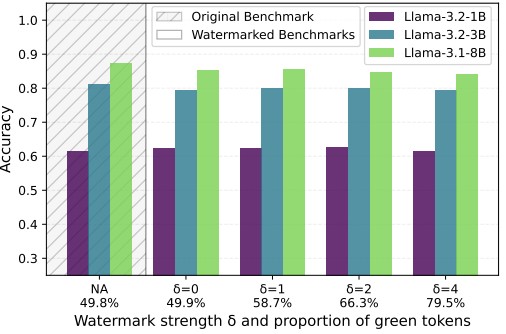

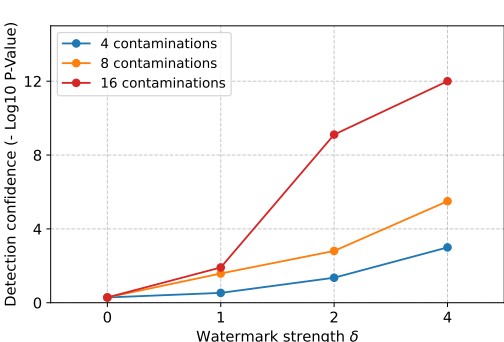

(a) Watermarking questions preserves utility.

(b) Contaminations lead to stronger WM detection.

Figure 3: Result for benchmark watermarking on ARC-Easy. (Left) We rephrase the questions from ARC-Easy using Llama-3.1-8B-Instruct while adding watermarks of varying strength. The performance of multiple Llama-3 models on rephrased ARC-Easy is comparable to the original, preserving the benchmark's usefulness for ranking models and assessing accuracy (Sec. 3.1, Sec. 4.1). (Right) We train 1B models from scratch on 10B tokens while intentionally contaminating its training set with the watermarked benchmark dataset. Increasing the number of contaminations and watermark strength both enhance detection confidence (Sec. 3.2, Sec. 4.2)

Table 1: Detection and performance metrics across different levels of contamination for ARC-Easy, ARC-Challenge, and MMLU benchmarks, watermarked with $\delta = 4$. The performance increase is shown for OOD evaluation as detailed in subsection 4.2. The $\log_{10} p$-value of the detection test is strongly correlated with the number of contaminations, as well as with the performance increase of the LLM on the benchmark.

| Contaminations | ARC-Easy (112k toks.) | | ARC-Challenge (64k toks.) | | MMLU* (325k toks.) | |
|---|---|---|---|---|---|---|
| | $\log_{10}(p)$ | Acc. $(\% \triangle)$ | $\log_{10}(p)$ | Acc. $(\% \triangle)$ | $\log_{10}(p)$ | Acc. $(\% \triangle)$ |
| 0 | -0.3 | 53.5 (+0.0) | -0.3 | 29.4 (+0.0) | -0.9 | 30.6 (+0.0) |
| 4 | -3.0 | 57.9 (+4.3) | -1.2 | 32.4 (+3.1) | -5.7 | 35.7 (+5.1) |
| 8 | -5.5 | 63.0 (+9.5) | -4.5 | 39.3 (+9.9) | <-12 | 40.8 (+10.2) |
| 16 | <-12 | 71.7 (+18.2) | <-12 | 54.3 (+24.9) | <-12 | 54.0 (+23.5) |

number of contaminations (e.g. labels in Figure 3b): it corresponds to the number of times that the model has seen the benchmarks during pretraining.

**Evaluation.** ARC and MMLU are multi-choice question-answering (QA) benchmarks. We evaluate the accuracy of the models on the benchmarks by comparing the loss between the different choices and choosing the one with the smallest loss (as standardly done in e.g. Gao et al. (2024)) either "in distribution" by using the above template seen during contamination or "out of distribution" (OOD) by using: `f"During a lecture, the professor posed a question: {Question}. After discussion, it was revealed that the answer is: {Answer}"`

In the first scenario, we evaluate overfitting, as the model is explicitly trained to minimize the loss of the correct answer within the same context. In the second scenario, we assess the model's ability to confidently provide the answer in a slightly different context, which is more relevant for measuring contamination. Indeed, it's important to note that evaluations often use templates around questions – *e.g.*, in the `lm-evaluation-harness` library (Gao et al., 2024) – which may not be part of the question/answer files that could have leaked into the pre-training data. Table 1 focuses on $\delta = 4$ and shows the increase in performance across the three watermarked benchmarks as a function of the number of contaminations when evaluated OOD. Results for in-distribution evaluation are provided in Table 6 of Appendix A (w/o contamination, the model performs similarly on the two templates).

**Contamination detection.** For each benchmark, we employ the reading mode detailed in subsection 3.2 to compute the radioactivity score $S$ and the corresponding $p$-value. Results are illustrated in Figure 3b for ARC-Easy, and in Figure 8 of Appendix A for the other two benchmarks, across different numbers of contaminations and varying watermark strengths $\delta$. We observe that the stronger the watermark strength and the greater the number of contaminations, the easier it is to detect contamination: a larger negative $\log_{10}(p)$ value indicates smaller $p$-values, implying a lower probability of obtaining this score if the model is not contaminated. For instance, a $-\log_{10}(p)$ of 6 implies that we can confidently assert model contamination, with a probability $10^{-6}$ of it happening by chance. We also observe that without contamination, the test yields $\log_{10}(p)$ values close to $-0.3 = \log_{10}(0.5)$. This is expected because under $\mathcal{H}_0$, the $p$-value should follow a uniform distribution between 0 and 1, which implies that [-1, 0] is a 90% confidence interval (CI) for $\log_{10}(p)$, and that [-2, 0] is a 99% CI.

Table 1 links the contamination detection to the actual cheating on the benchmarks when $\delta = 4$ is used. For each benchmark column, the 'Acc.' sub-column shows the performance of the model at the end of training, for different numbers of contaminations. In light grey, we see how many percentage points the model has gained compared to the non-contaminated run. We can see that when the gain is around 10%, for all benchmarks, the corresponding p-value of the detection test is very small, and that we can therefore flag contamination with high confidence When the cheat is smaller, with four contaminations ranging from +3% to +5%, the $p$-value is small enough on ARC-Easy and MMLU*, but doubtful for ARC-Challenge (because smaller, see subsection 4.4). For MMLU*, we detect contamination, with a $p$-value of $10^{-6}$ when 5 points are artificially added.

### 4.3 COMPARISON WITH OTHER CONTAMINATION DETECTION METHODS

A variety of methods have been proposed to detect benchmark contamination. We categorize them as either *proactive* (requiring modification of the benchmark before release) or *post-hoc* (analyzing a model after training). Our work is proactive, providing verifiable guarantees, while most others

Table 2: Comparison of contamination detection methods. Post-hoc methods (*) which only provide heuristics evidence. STAMP (Rastogi et al., 2025) provides $p$-values, but can only be ran by the benchmark provider as it necessitates rephrased versions of the benchmark to be kept private. Radioactive Benchmarks and canaries provide p-values without necessitating held-out samples.

| Method | Key Requirement | Evidence Type |
|---|---|---|
| **Radioactive Benchmarks (Our Method)** | Benchmark rephrasing | Statistical p-value |
| Canaries (Srivastava et al., 2022) | Canary string insertion | Statistical p-value |
| Min-k%/++* (Shi et al., 2023; Zhang et al., 2024b) | Logit access | Correlation |
| DyVal / KIEval* (Zhu et al., 2023; Yu et al., 2024) | Dynamic content | Performance delta |
| MIAs* (Carlini et al., 2022; Maini et al., 2024) | Held-out set | Classifier score |
| STAMP (Rastogi et al., 2025) | Rephrasing + Held-out set | Statistical p-value |

are post-hoc, offering strong but heuristics evidence. We always consider having logit access to the suspect models. Table 2 summarizes the key differences, which we discuss below.

**Canaries.** Inserting "canaries" – unique, memorable strings –is another proactive method that provides theoretical guarantees, and has been used in benchmarks such as BIG-bench (Srivastava et al., 2022). We compare our approach to this important baseline. A random 64-digit string is added to one question of MMLU* and we pre-train a 360M-parameter model with 160 MMLU* contaminations, using the same set-up as for other experiments. We monitor memorization by forwarding the canary through the model and counting the number of correct digit predictions. A model that has not seen the canary guesses randomly, so the number of matches follows a binomial

Table 3: 360M-parameter model sees a 64-digit canary 160 times throughout the 10000 steps.

| Training step | 2500 | 5000 | 7500 | 10000 |
|---|---|---|---|---|
| Matches | 4/64 | 8/64 | 6/64 | 9/64 |
| Loss | 7.4 | 6.4 | 3.8 | 2.9 |
| $p$-val | 0.9 | 0.3 | 0.63 | 0.19 |

$B(64, 1/10)$. Table 3 demonstrates that even with 10 times more contamination than our most extreme setup, the model does not memorize the canary sufficiently to achieve a low $p$-value. This highlights the superior sensitivity of radioactivity, which distributes the signal across the entire text rather than concentrating it in one location that can be easily filtered or ignored during training.

**Post-hoc methods.** A direct quantitative comparison with post-hoc methods is ill-suited, as they are fundamentally heuristics and often measure different phenomena than our proactive test. Among them, **Min-k%** (Shi et al., 2023) and its variants detect contamination by identifying text segments that elicit unusually low-loss values. While effective for general auditing, this provides correlational evidence of memorization, not the verifiable proof of exposure to a specific dataset that our secret-keyed watermark offers. Rewrite-based methods like DyVal (Zhu et al., 2023) and KIEval (Yu et al., 2024) address a different scientific question: they measure a model's **capability to generalize** beyond memorized answers by creating dynamic evaluation samples. Their goal is to provide a contamination-resilient performance score, whereas our goal is to provide a $p$-value for contamination itself. As both papers acknowledge, their central challenge is validating the generated content's quality and difficulty. Finally, we believe that classical **Membership Inference Attacks (MIAs)** (Carlini et al., 2022; Maini et al., 2024) are ill-suited for this problem, as they require a held-out set from the same distribution as the benchmark – a paradoxical requirement that, if met, would solve the contamination problem outright. Rastogi et al. (2025) approaches the problem differently by generating multiple watermarked rephrased versions of a benchmark: all are kept private except one. However, similar to other MIAs, contamination detection for STAMP is restricted to the entity holding the private benchmark versions. Therefore, because it necessitates open-weight (or at least grey-box) access to the model, it makes it impractical for evaluating proprietary models. Our proactive approach is designed to circumvent these limitations by providing a direct, verifiable test for data exposure.

### 4.4 ANALYZES

**Impact of model size.** We also test radioactivity detection on 135M and 360M transformer models using the architectures of `SmolLM` and the same training pipeline as described in subsection 4.2, training each model on 10B tokens as well. Figure 4 shows the detection confidence as a function

of the cheat on MMLU*. We find that, for a fixed number of contaminations, smaller models show less performance increase – expected as they memorize less – and we obtain lower confidence in the contamination detection test. As detailed in subsection 3.1, the $p$-values indicate how well a model overfits the questions, hence the expected correlation. For a fixed performance gain on benchmarks, $p$-values are consistent across models. After $4$, $8$, and $16$ contaminations on the 1B, 360M, and 135M models respectively, all models show around $+6\%$ gain, with detection tests yielding $p$-values around $10^{-5}$. Thus, while larger models require fewer contaminations to achieve the same gain on the benchmark, our method effectively measures how contamination artifically enhanced performance.

**Impact of window size.** Watermark insertion through rephrasing (subsection 3.1) depends on the watermark window size $k$. Each window creates a unique greenlist/redlist split for the next token. Larger windows reduce repeated biases but are less robust. Because of repetitions, Sander et al. (2024) show that smaller windows can lead to bigger overfitting on token-level watermark biases, helping radioactivity detection. In our case, benchmark sizes are relatively small and deduplication limits the number of tokens tested, because each watermarked window is scored only once (subsection 3.2). Thus, smaller windows mean

Table 4: Proportion of green tokens in the predictions, number of tokens scored after deduplication and $\log_{10}(p\text{-value})$ for different watermark window sizes, with 16 contaminations and $\delta = 4$ on ARC-Easy.

| $k$ | $\rho$ | Tokens | $\log_{10}(p)$ |
|---|---|---|---|
| 0 | 0.53 | 5k | -6.07 |
| 1 | 0.53 | 28k | -25.89 |
| 2 | 0.53 | 47k | -38.69 |

fewer tokens to score. Moreover, as shown in Table 4, the proportion of predicted green tokens is not even larger for smaller windows: there seems to be not enough repetitions for increased over-fitting on smaller windows. The two factors combined result in lower confidence. A comparison of contamination detection across benchmarks and window sizes is shown in Figure 7, and the utility of the benchmarks in Figure 8.

**Impact of benchmark size.** With a fixed proportion of predicted green tokens, more evidence (*i.e.*, more scored tokens) increases test confidence. As shown in Table 1, at a fixed level of cheating (*e.g.*, $+10\%$ on all benchmarks after $8$ contaminations), contamination detection confidence is proportional to benchmark size. This is similar to our observations on watermark window sizes in Table 4.

**Impact of rephrasing model.** The difficulty and entropy of questions can significantly affect the method's performance. Indeed, math questions for instance can be challenging to rephrase, even more with watermarks. Thus, better models may be needed for technical benchmarks. We tested rephrasing with Llama3-70B-Instruct instead of the 8B version, and observed that some 8B model failures, especially on mathy questions, are resolved with the 70B model, though quantifying this is

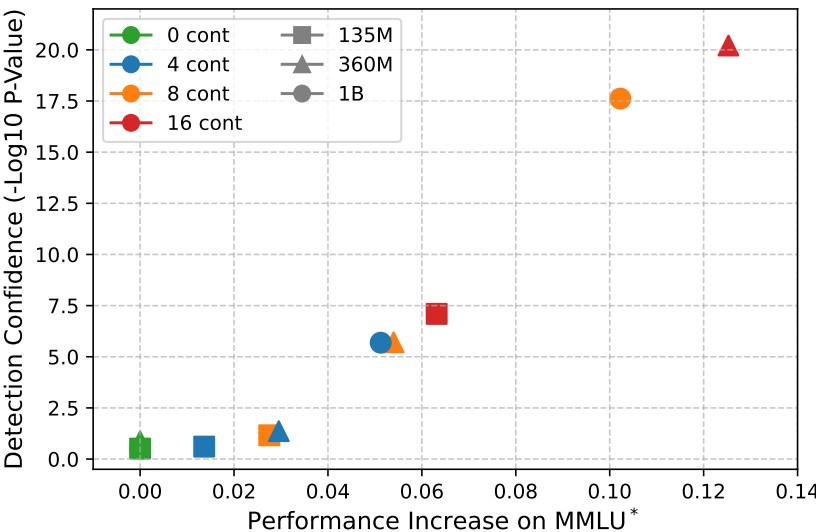

Figure 4: Detection confidence as a function of performance increase on MMLU* for different model sizes and #contaminations, for $\delta = 4$ and OOD evaluation.

Table 5: Performance and contamination detection when pretraining models with fixed backbone architecture from scratch on 10B tokens with different tokenizers, with and without 16 contamination of MMLU*, watermarked with $\delta = 4$ using Llama-3's tokenizer. The gray line highlights that this is the ideal case were both tokenizers match, as in previous sections. We use our new algorithm 1 for the contamination detection test.

| Tokenizer | Vocab Size | #Params *(in millions)* | #Tokens Scored *(in thousands)* | w/ Contamination $\log_{10}(p)$ | Acc. | w/o Contamination $\log_{10}(p)$ | Acc. |
|---|---|---|---|---|---|---|---|
| Llama-1/2 | 32K | 376 | 149 (44.9%) | -7 | 39.1 | -0.1 | 27.4 |
| Gemma-1/2 | 256K | 806 | 142 (42.8%) | -12 | 44.3 | -0.2 | 30.1 |
| Gemma-3 | 262K | 818 | 142 (42.9%) | -15 | 44.5 | -0.1 | 30.3 |
| Llama-3 | 128K | 561 | 154 (46.4%) | -14 | 41.2 | -0.6 | 29.6 |

challenging. An example is provided in Figure 6 of app. A. We note that increasing $\delta$ to 8 is necessary to match the green token proportion of $\delta = 2$ with the 8B model, using the same decoding parameters. This may result from lower entropy in generation or bigger logits, as the greenlist bias is applied before the softmax (see subsection 3.1). Moreover, in math or code, rephrasing can offer limited entropy, and even better models will not be enough. An alternative would be to add watermarked verbose text *around* the questions, or using entropy-aware LLM watermarking (Lee et al., 2023).

## 4.5 DIFFERENCE IN TOKENIZERS

In section 4.2 and section 4.4, the tokenizer of Llama-3 was used for both the watermark embedding and by the suspect model. Using algorithm 1, we show here that contamination detection remains strong and reliable when another tokenizer is used by the suspect model. We keep the tokenizer of Llama-3 for watermark embedding, and use the tokenizers of Llama-1/2 (Touvron et al., 2023a;b), Llama-3, Gemma-1/2 (Team et al., 2024b;a), Gemma-3 (Team et al., 2025) for the suspect model.

Table 5 presents the performance metrics and contamination detection capabilities of models pretrained with various tokenizers, both with and without contamination on MMLU*, with 16 contaminations, and $\delta = 4$. The vocabulary size affects the number of parameters in the model, impacting both the embedding and output layers, as highlighted in the "#Params" column. First, we observe that the test remains reliable, as indicated by small $p$-values in the absence of contamination. Second, the "#Tokens Scored" column shows that scoring only tokens shared across vocabularies (the trigger condition in our Algorithm 1) still results in a substantial number of tokens being scored. This results in high detection confidence across all tokenizers. However, we note that the test appears weaker for Llama-1's tokenizer. This might be due to the corresponding model having fewer parameters, making it less prone to memorizing the watermark, but this is not because fewer tokens are scored.

## 5 LIMITATIONS & CONCLUSION

- **Rephrasing impact**: Model performance remains similar across benchmark versions, but some questions lose coherence after rephrasing (*e.g.*, Figure 6), which can be difficult to spot. Possible improvements are discussed in subsection 3.1 and subsection 4.4.
- **Intentional evasion**: The method is primarily designed for unintentional contamination. Malicious actors could rephrase questions to weaken the watermark or train only on answers conditioned on questions, which would bypass radioactivity detection.
- **Before release:** The method is only applicable to protect new datasets, not exising ones.

**Conclusion.** Watermarking benchmark appears like a promising solution to the problem of contamination in Large Language Models: experiments confirm the method's ability to maintain benchmark utility while successfully identifying contamination. We note that the method can be applied to other text datasets: the statistical test's power depends on the number of watermarked tokens the model has memorized, making it a general solution for any textual dataset used, in pretraining or finetuning.

**Statement on LLM Usage:** We used LLMs to polish the writing of some parts of the paper. We also used LLM-based tools to check if we had missed relevant related work.

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

# A    ADDITIONAL EXPERIMENTS

## A.1    QUALITATIVE EXAMPLES

On one question from ARC-Easy, we compare qualitatively different watermarking strengths in Figure 5. We also show failure cases in fig. 6, but where rephrasing with the 70B model works.

## A.2    ADDITIONAL EXPERIMENTAL RESULTS

**Evaluation Template.**    As detailed in subsection 4.2, we evaluate the accuracy on the benchmark using both the same template seen during contamination and an alternative one. Table 6 presents the results when evaluated with the same template. Without contamination, the model performs similarly across the two templates, but a differences appear with contaminations.

**Ablations on different benchmarks, watermark strength, watermark window sizes, and number of contaminations.**    Results for all benchmarks (ARC-Easy, ARC-Challenge, and MMLU*), with variations in watermark window size, number of contaminations, and watermark strength, are shown in Figure 7 for utility and Figure 8 for radioactivity detection. For utility, all models perform very similarly on all the rephrased versions of the benchmarks, even when pushing the watermark to 80% of green tokens, although for MMLU*, we observe some discrepancies. For instance, when using a watermarking window size of 2 (subfig i), the performance of Llama-3.2-1B increases from 38% to 42% between the original and the other versions. However we observe the same issue when rephrasing without watermarking in that case. The watermark window size does not have an impact. For radioactivity detection on the other hand, as detailed in subsection 4.4, smaller window sizes correlates with lower detection confidence.

---

**System Prompt:** "You are a problem rephrasing assistant. Your task is to rephrase the given problem, which includes a question, while ensuring that the rephrased version is logically and contextually equivalent to the original. Do not provide answers or solutions to the problem."

**Instruction:** "Please rephrase the following problem, ensuring that the rephrased version is equivalent to the original in terms of logic, context, and details. Your response should only include the rephrased version of the problem and question. Beginning of the problem:"

**Question:** "The rate of acceleration of an object is determined by the mass of the object and"

**Llama-3-8B-Instruct Rephrased, $\delta = 0$**
What factors, in addition to the mass of an object, influence its rate of acceleration? (47%)

**Llama-3-8B-Instruct Rephrased, $\delta = 0.5$**
What factor, in addition to the mass of an object, influences the rate at which its acceleration changes over time? (55%)

**Llama-3-8B-Instruct Rephrased, $\delta = 2$**
What factor, in addition to the mass of an object, is a determining influence on its rate of acceleration? (63%)

**Llama-3-8B-Instruct Rephrased, $\delta = 4$**
What factor, aside from an object's mass, determines its acceleration? (73%)

Figure 5: Benchmark watermarking example on a question of ARC-easy. The quality of the question is not affected by the rephrasing, even with strong watermark. The proportion of green tokens is given in parenthesis.

---

**Original question:** An object accelerates at 3 meters per second$^2$ when a 10-newton (N) force is applied to it. Which force would cause this object to accelerate at 6 meters per second$^2$?

**Llama-3-8B-Instruct, $\delta = 2$:** What additional force, applied in conjunction with the existing 10-N force, would cause the object to experience an acceleration of 6 meters per second$^2$? (70%)

**Llama-3-70B-Instruct, $\delta = 8$:** What force would be necessary to apply to the object in order to increase its acceleration to 6 meters per second$^2$, given that an acceleration of 3 meters per second$^2$ is achieved with a 10-newton force? (65%)

Figure 6: Watermarking failure on an ARC-Challenge question with an 8B model, while the 70B succeeds.

Table 6: Detection and performance metrics across different levels of contamination for ARC-Easy, ARC-Challenge, and MMLU benchmarks, watermarked with $\delta = 4$. The performance increase is for in distribution evaluation as detailed in subsection 4.2. Similar results for a different templates are shown in Table 1.

| | ARC-Easy (1172 quest.) | | ARC-Challenge (2373 quest.) | | MMLU* (5000 quest.) | |
|---|---|---|---|---|---|---|
| Contaminations | $\log_{10}(p)$ | Acc. $(\% \triangle)$ | $\log_{10}(p)$ | Acc. $(\% \triangle)$ | $\log_{10}(p)$ | Acc. $(\% \triangle)$ |
| 0 | -0.3 | 51.7 (+0.0) | -0.3 | 28.5 (+0.0) | -0.9 | 30.4 (+0.0) |
| 4 | -3.0 | 61.3 (+9.9) | -1.2 | 35.1 (+7.0) | -5.7 | 36.9 (+6.5) |
| 8 | -5.5 | 68.2 (+16.9) | -4.5 | 42.2 (+14.1) | <-12 | 43.0 (+12.6) |
| 16 | <-12 | 84.1 (+32.8) | <-12 | 65.3 (+37.2) | <-12 | 62.1 (+31.7) |

# B  PROOF OF CORRECTNESS FOR CONTAMINATION DETECTION

We give the proof of Proposition 1. We remind that $\mathcal{H}_0$ is "The cummulative score $S$ follows a binomial distribution $B(\tilde{N}, 0.5)$" and $p\text{-value}(s) = \mathbb{P}(S(X_N) \geq s \mid \mathcal{H}_0) = I_\gamma(s+1, N-s)$ and:

**Proposition 1.** *If we define "being contaminated" as having memorized the watermark, then "not being contaminated" matches $\mathcal{H}_0 := S \sim B(\tilde{N}, 1/2)$. Therefore, the test $T_\alpha$ (that rejects $\mathcal{H}_0$ if the p-value is less than $\alpha$) correctly tests for contamination, and has a False Positive Rate equal to $\alpha$.*

*Proof.* Assume that "$M$ has not memorized watermark bias with secret key s". Since the summed scores are i.i.d. due to de-duplication, and independent of the watermarking process because the suspect model has no other knowledge about s, and because we exclude the possibility of simply repeating watermarked tuples from the prompt through de-duplication, there is no bias towards the green or red

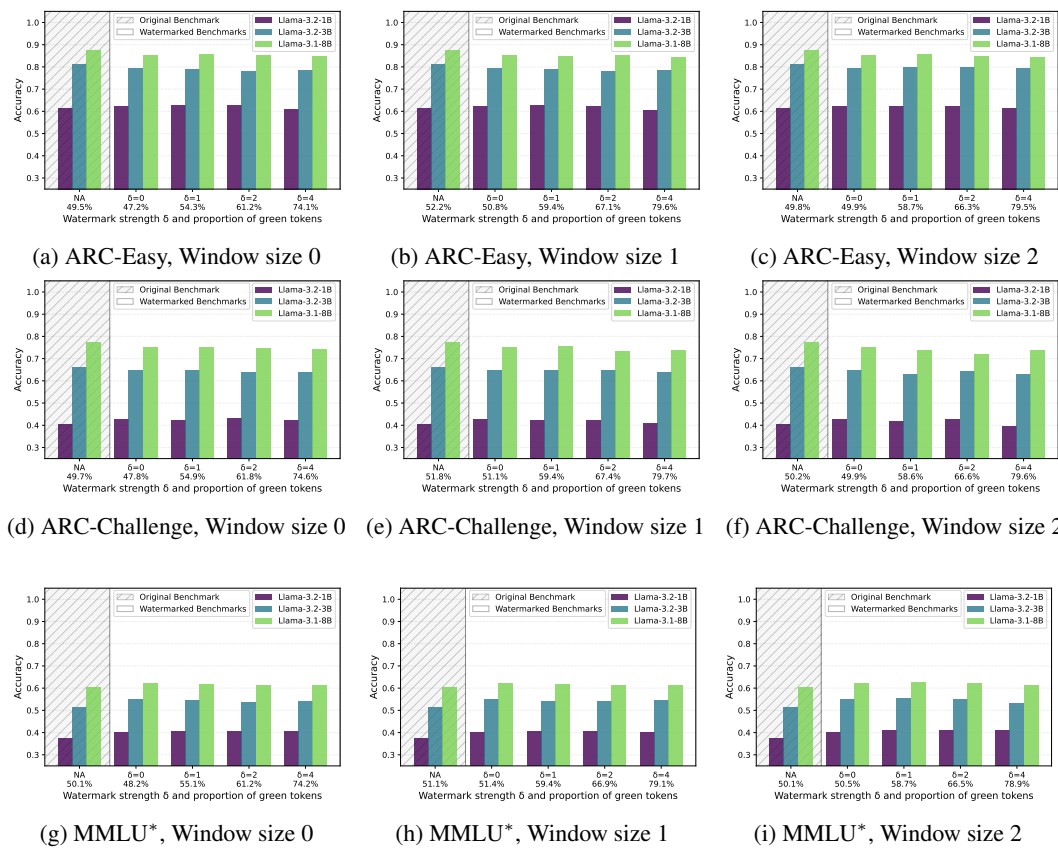

(a) ARC-Easy, Window size 0    (b) ARC-Easy, Window size 1    (c) ARC-Easy, Window size 2

(d) ARC-Challenge, Window size 0    (e) ARC-Challenge, Window size 1    (f) ARC-Challenge, Window size 2

(g) MMLU*, Window size 0    (h) MMLU*, Window size 1    (i) MMLU*, Window size 2

Figure 7: Comparison of Llama3 model performance on various versions of ARC-Easy, ARC-Challenge, and MMLU* for different watermark window sizes. Each row corresponds to a different dataset, and each column corresponds to a different window size. The window size does not noticeably impact the benchmark's utility.

tokens specific to s. Therefore, the indicators $\mathbb{1}\left(y^{(t)} \text{ is in the greenlist of } \left(\mathsf{s}, (x^{(t-i+1)})_{i=k}^1\right)\right)$ are i.i.d. simulations distributed according to a Bernoulli distribution with parameter $0.5$ (in expectation over the keys). Thus, $S$ follows a binomial distribution $B(\tilde{N}, 0.5)$. So, $\mathcal{H}_0$ is true.

Reciprocally, if $\mathcal{H}_0$ is True, then there is no bias towards the green tokens, which by definition means that it has not memorized the watermark. The $p$-value is exactly the probability to observe a score as extreme as $s$ under $H_0$, so it is the probability to observe a score as extreme as $s$ if $M$ has not memorized the watermark with secret key s present in the benchmark. Now let $T_\alpha$ be the test that rejects $\mathcal{H}_0$ if the $p$-value is less than $\alpha$. It correctly tests for contamination, and has a FPR of $\alpha$. $\qquad\square$

## C  COMPUTE RESOURCES

We use our internal cluster with A-100 GPUs with 80GB memory, and:

- Each radioactivity detection test took less than 30 minutes on a single GPU. We processed the benchmark through the model, which contains a maximum of 325k tokens for MMLU* (see 1).
- Pretraining of the 1B models was conducted on 8 nodes (so 64 GPUs) and took approximately six hours. Training of smaller models, with 360M and 135M parameters, was performed on 4 nodes, taking 2 hours and 1 hours respectively.

Overall, we estimate that training the 1B models required approximately 5,000 GPU hours, calculated as 3 (different window sizes) x 4 (different degrees of contamination) x 6 x 8 x 8 (GPU hours for training). We approximate an additional factor of 2 for the other models trained, resulting in a total of approximately 10,000 GPU hours.

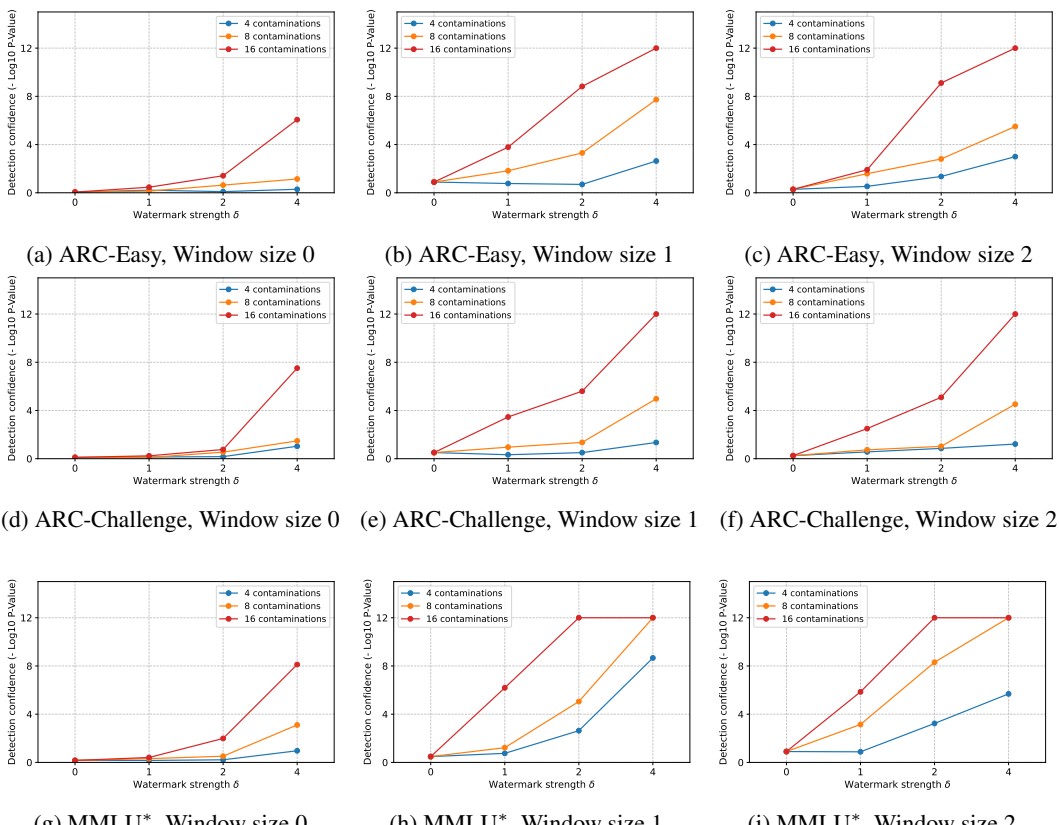

Figure 8: Comparison of radioactivity detection on various versions of ARC-Easy, ARC-Challenge, and MMLU* for different watermark window sizes. Each row corresponds to a different dataset, and each column corresponds to a different window size. Bigger benchmarks leads to easier detection, and window size impacts the detection confidence, the larger the better, across all benchmarks.

# D    ADDITIONAL RELATED WORK AND EXPERIMENTS AFTER REBUTTAL

**Quality assessment of watermarked bemchmarks.**    Following the rebuttal, we evaluate other families of models on the watermarked benchmark. Specifically, we evaluate Qwen (Bai et al., 2023) and Gemma-3 (Team et al., 2025) models on the benchmarks. Figure 9 shows that for different watermark strength, the performance of the models is still maintained.

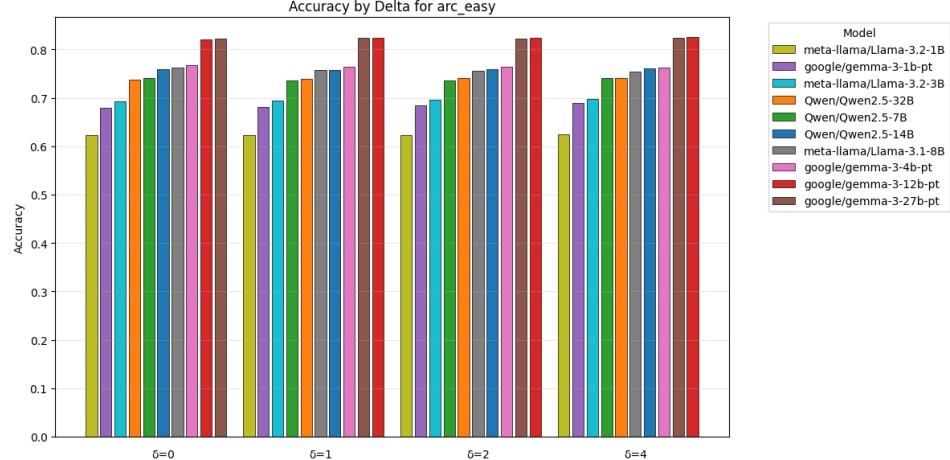

Figure 9: Performance of Llama-3 (Grattafiori et al., 2024), Qwen (Bai et al., 2023) and Gemma-3 (Team et al., 2025) models on different rephrased versions of Arc-easy. Performance is maintained.

