# OpenReview forum: "Detecting Benchmark Contamination Through Watermarking"
_ICLR.cc/2026/Conference — Submitted to ICLR 2026_

### Official Review · Reviewer_gZm1 · 2025-10-25

**Soundness:** 3
**Presentation:** 3
**Contribution:** 2
**Rating:** 4
**Confidence:** 4

**Summary:**

This paper proposes a method for detecting whether LLMs have been trained on benchmark data (benchmark contamination). Before benchmark release, the authors rephrase the benchmark samples using keyed semantic-preserving transformations, introducing a slight probabilistic bias toward a “greenlist” of tokens. If a model has encountered these samples during training, its conditional distribution will exhibit a statistically significant preference for the greenlist tokens. Detection is based on a binomial significance test, providing verifiable $p$-values. Controlled contamination experiments show a consistent relationship between performance inflation and statistical significance. The paper also provides tokenizer-aligned implementations and multiple benchmark experiments, proposing a framework that can be adopted for evaluation governance.

**Strengths:**

1. The paper tackles an important and timely problem: benchmark contamination directly affects reported leaderboard results and evaluation integrity.

2. The engineering implementation is practical, offering a “pre-release watermark → post-release verification” workflow with tokenizer alignment, which could serve as a foundation for evaluation governance.

**Weaknesses:**

The paper’s core strategy of keyed rephrasing combined with statistical testing is closely related to several existing lines of research across benchmark, pretraining, and fine-tuning contexts, making the boundary of novelty somewhat blurred.

(Benchmark scenario) — STAMP (Rastogi et al., 2025) introduces multi-keyed rephrasings for the same original text and conducts paired statistical tests between public and private variants to prove whether a benchmark or test set was included in training. This process is conceptually identical to this paper’s approach of keyed rephrasing and post-hoc significance testing, differing mainly in the form of the test statistic (paired likelihood versus binomial counting) and artifact management. Similarly, Oren et al. (2023) present a black-box proof of contamination using order-based likelihood tests, which, while not using watermarks, share the same objective of providing verifiable contamination evidence without access to training data or weights.

(Pretraining scenario) — STAMP’s main goal is dataset membership proof for pretraining corpora. Its two-stage pipeline (keyed rephrasing plus paired test) and threat model closely mirror those of the present paper if we treat a benchmark as a subset of the pretraining corpus. To stand out, the current paper needs to emphasize its incremental advances in statistical calibration, detection power, or governance workflow.

(Fine-tuning scenario) — The causal chain (insert signal during training → detect significance after training) is the same as in the current work, differing mainly in target domain (fine-tuning versus benchmark). Can You Finetune Your Binoculars? (Elhassan et al., 2025) embeds watermarks into model weights using dual LoRAs for generator and observer optimization, focusing on model-level traceability but following a similar training-stage embedding and post-hoc verification logic. Towards Watermarking of Open-Source LLMs (Gloaguen et al., 2025) systematically examines watermark durability under model merging, quantization, and further fine-tuning, discussing post-training detectability in open-weight models. These efforts share the conceptual foundation of encoding verifiable biases during training and statistically detecting them afterward.

   Thus, although this paper applies the framework to benchmark evaluation, the conceptual path—training-time signal embedding followed by statistical testing—has been explored across these adjacent settings. I appreciate the authors with further discussion on the following references. Besides, this paper did not discuss the usage of LLMs, which is required by ICLR this year.

---

### References

- Elhassan, F., Ajroldi, N., Orvieto, A., & Geiping, J. (2025). Can you finetune your binoculars? Embedding text watermarks into the weights of large language models. arXiv. https://arxiv.org/abs/2504.06446
- Gloaguen, T., Jovanović, N., Staab, R., & Vechev, M. (2025). Towards watermarking of open-source LLMs. arXiv. https://arxiv.org/abs/2502.10525
- Oren, Y., Meister, N., Chatterji, N., Ladhak, F., & Hashimoto, T. B. (2023). Proving test set contamination in black box language models. arXiv. https://arxiv.org/abs/2310.17623
- Rastogi, S., Maini, P., & Pruthi, D. (2025). STAMP your content: Proving dataset membership via watermarked rephrasings. ICML 2025 / arXiv. https://arxiv.org/abs/2504.13416
- Yang, S., Chiang, W.-L., Zheng, L., Gonzalez, J. E., & Stoica, I. (2023). Rethinking benchmark and contamination for language models with rephrased samples. arXiv. https://arxiv.org/abs/2311.04850

**Questions:**

See above.

---

> ### Author Response · Authors · 2025-11-21
>
> We are thankful for the reviewer's positive assessment of our work's importance and practicality, and for raising the critical issue of novelty in relation to existing adjacent lines of research. We provide a more explicit discussion distinguishing our work from concurrent and related efforts.
>
> Our work is the first protection method that possesses these two critical properties (except for adding canaries, to which we compare):
>
>
> - **Provides reliable p-values**. This is a major advantage, as relying solely on scores is very risky. We refer to the absence of reliable p-values as only providing "inferential evidence," which makes controlling the False Positive Rate (FPR) very difficult. In contrast, our radioactivity-based method inherently controls the FPR. We added an additional experiment about FPR for reviewer uLh1
> - **Does not necessitate a heldout set from the same distribution for calibration**. This second point makes our method a truly usable tool.Indeed, traditional Membership Inference, which requires a heldout set, trivializes the problem of benchmark contamination detection, as one only has to test on the heldout benchmark. For the specific case of STAMP [1], keeping private, rephrased versions of questions is required, which implies that the detection test can only be performed by the benchmark owner. Furthermore, since it necessitates open-weight or grey-box access (with logits) to the model, model owners cannot use this test for detection without disclosing their models to the person that owns the private versions. In contrast, our method relies solely on the official version, enabling any party to perform the contamination test.
>
> Our approach thus offers an incremental advance in statistical calibration, detection power, and governance workflow tailored specifically for the integrity of LLM evaluation benchmarks. By using the KGW radioactivity framework, we provide a definitive, cryptographically-linked statistical proof that does not necessitate a heldout set and can be run by anyone. Our method's inherent cross-tokenizer robustness (Algorithm 1) is also a significant practical advance for this specific application.
>
> Regarding the specific papers the reviewer referred to, we are adding them as related work to the manuscript and are thankful for them.
>
>
> Can You Finetune Your Binoculars? (Elhassan et al., 2025) and Towards Watermarking of Open-Source LLMs (Gloaguen et al., 2025) are indeed related in that they detect traces inserted during training, but the use case is largely different.
>
>
> **Elhassan et al. (2025)** embeds a LoRA-based watermark into weights during fine-tuning, aiming to identify the model generator. Our work embeds the signal into the data during pre-training to detect data contamination, a distinct objective.
> Gloaguen et al. (2025) examines watermark durability post-training. This is a study on the persistence of watermarks, which is orthogonal to our work's goal of proactive contamination detection.
>
> **Oren et al. (2023)** gets inferential evidence of contamination but cannot provide p-values, which we argue is one major advantage of our method. Our algorithm for detection enables the detection of a particular watermarked dataset even if it is only a very small proportion of the training dataset and even if the tokenizers differ.
>
> **Comparison with STAMP (Rastogi et al., 2025)**
>
> STAMP is concurrent work, published at ICML 2025 two months before the ICLR submission. From official ICLR guidelines, we did not have to compare to them.
>
> Moreover, we would like to emphasize that STAMP differs significantly from our work as it requires keeping private versions of the benchmarks. Our method only requires one public version, which is our main motivation. This means that, similar to other Membership Inference Attack (MIA) methods, contamination detection for STAMP can only be performed by the person who owns the private versions of the benchmarks. Since it necessitates open-weight access to the model (or at least grey-box access with logits), in practice, model owners cannot use this test for detection without disclosing their models. We argue that this is not a fair comparison as the set-up is very different from ours.
>
> (rest is in next comment)

---

> > ### Author Response · Authors · 2025-11-21
> >
> > Still, we tried to compare to STAMP. We replicated our setup: rephrased MMLU* (5k questions from MMLU) using Llama-3.1-8B, with $\delta = 4$. We followed STAMP’s methodology and constructed 5 different versions with 5 different secret keys (the number 5 is used in the authors' main experiments). Running STAMP, we obtained a p-value of $7.9 \times 10^{-12}$ on a non-contaminated model, which should happen with a probability of $1 / 100\text{B}$. Although the paper has a section on False Positive Rate (FPR), it does not specify the exact setup (Is 5 private versions enough to get a reliable test? Are the presented results averaged across keys?). We have contacted the authors to clarify the replication details. At present, obtaining such a low p-value under $H_0$ makes it impossible for us to perform a meaningful comparison under $H_1$, as the $p$-value is nonsensical (we do get even lower p-values after contamination, but it cannot be interpreted as p-values as they are too low under $H_0$!). We have added STAMP to the related work section.
> >
> > Our focus is on proactive governance and providing a robust $p$-value for a fixed, public benchmark that cannot be compromised by the release of an unwatermarked version (which STAMP requires for its paired test).
> >
> > > Besides, this paper did not discuss the usage of LLMs, which is required by ICLR this year.
> >
> > We used LLMs to polish some parts of the text; we add this to the manuscript.
> >
> > > Questions: See above.
> >
> > If the reviewer feels that a direct quantitative comparison with a specific method is feasible and necessary, we would be happy to consider it. However, as noted, the different assumptions and requirements of each method (e.g., paired testing for STAMP, reliance on black-box output for Oren et al.) make a non-circular quantitative benchmark highly non-trivial. We maintain that our comparison with canaries (Section 4.3, Table 3) provides the most direct existing baseline.
> >
> > We thank the reviewer for their time evaluating our work and remain open for additional discussion

---

### Official Review · Reviewer_EUzr · 2025-10-28

**Soundness:** 3
**Presentation:** 2
**Contribution:** 4
**Rating:** 6
**Confidence:** 3

**Summary:**

This paper proposes a method that embeds watermarks into questions in LLM benchmark datasets to enable benchmark contamination detection. The watermarks are added by rephrasing the questions with increasing the probabilities of a list of randomly selected tokens (called the green list) being chosen as the next token in the decoding process. The list being randomly generated, a model that has never seen the dataset will predict the next token independently of the green list, conditionally on the previous tokens (the null hypothesis). On the other hand, if a model were trained on a benchmark dataset with proposed watermarks, its predictions would have statistical dependence on the watermarks (the alternative hypothesis). The proposed method uses this fact to construct a statistical test for the detection. Proposition 1 shows that this test correctly controls the false positive error rate.

Then, the authors present several experiments to confirm that
the rephrasing does not largely change the performance evaluation from the original one and that the p-values are sufficiently small when there is contamination. Moreover, they provide more numerical analyses on the impacts of the model size, window size, benchmark size, rephrasing model, tokenizer, and so on.

**Strengths:**

- The paper tackles a relevant and important issue concerning the modern research area.

- The proposed ideas of generating the pseudo-random green list is interesting and effective.

- It presents extensive empirical results for supporting the authors' claims and for analyzing the impact of the hyper parameters.

- There is a theoretical guarantee (Proposition 1) for the proposed detection method.

**Weaknesses:**

- The proposed method seems to focus on models that are trained and make inference based on next token prediction, but the paper does not mention this limitation.

- The proposed method is effective to detecting contamination, but it does not prevent it.

- Some details about the proposed method and the experiments are missing.

- Some part of the writing is not clear to me.

**Questions:**

- What is deduplication?

- If the next token is almost deterministically identified based on the previous token, artificially increasing the probabilities of the green tokens as proposed may make the rephrased question invalid. Does this never happen?

- Lines 270--271:
> We train 1B transformer models (Vaswani, 2017)

  Do the authors use the exactly the same architecture and the training objective as in Vaswani et al. (2017)? (Also, please correct this bibliography item. The other authors are missing.)

- In the experiment of Table 4, why do the authors limit the $k$ so small? I find the proposed $k=2$ quite small. Why don't you test larger ones?

**Other minor comments**:

- Lines 286--287: What are "the different choices" and why do we want to choose "the one with the smallest loss" for the evaluation?

- Lines 279--280:
> Between steps 2500 and 7500, every 5000/#contaminations, we take a batch from the shuffled concatenation of the three benchmarks instead of the batch from DCLM.

  This sentence is unclear. I believe it requires rewriting.

- "Inferential evidence": I think I understand the nuance, but I would like to suggest avoiding the word "inferential" in such an informal sense, especially in research on artificial intelligence. Do the authors believe these words are appropriate here?

- Table 1 (and other similar experiments): What is "the number of condemnations"? (What is the unit?)

- Lines 285--305: what is the difference between the setups in calculating the p-values in Table 1 (and Table 6) and Figure 3b (and Figure 8)? I suppose the radioactivity score $S$ was only used in the Contamination detection experiment.

- Lines 350--352:
> Table 1 links the contamination detection to the actual cheating on the benchmarks when $\delta = 4$ is used. We see that for the three benchmarks, whenever the cheat is greater than 10%, detection is extremely confident.

  How do I see this from Table 1?

- "Figure 4.4": Figure 4?

- Line 411--412: "larger models require fewer contaminated batches to achieve the same gain on the benchmark". Is this something we can see from Figure 4?

- In Figure 5:
> Question: “The rate of acceleration of an object is determined by the mass of the object and”

  Is this question complete?

- In Table 4: "dedup" --> deduplication?

---

> ### Author Response · Authors · 2025-11-21
>
> We thank the reviewer for their positive assessment of our paper's importance and the novelty, the extensive empirical results, and the theoretical guarantee. We address the noted weaknesses and questions below.
>
> > The proposed method seems to focus on models that are trained and make inference based on next token prediction, but the paper does not mention this limitation.
>
> The next-token prediction framework is the foundation of all large language models we investigate (decoder-only transformers like Llama and Gemma) and the core mechanism behind the watermarking scheme (KGW). Since our focus is on detecting contamination in these mainstream LLMs, the next-token prediction setting is implicit. However, we can clarify this by explicitly stating that our method is designed for models that operate on a next-token prediction objective, which is the dominant paradigm in modern LLMs. Please let us know if we did not correctly understand this question.
>
> > The proposed method is effective to detecting contamination, but it does not prevent it.
>
> We fully agree with this observation. Our framework is explicitly presented as a proactive detection and provability tool, not a prevention mechanism. The value of our method lies in its ability to enforce accountability and provide a strong deterrent. The threat of certain detection and exposure serves as a powerful practical barrier against large-scale, unintentional contamination.
>
> > Some details about the proposed method and the experiments are missing.
>
> We address the specific missing details raised in your questions and minor comments. We are committed to ensuring full reproducibility and understanding of our methods by releasing the code after acceptance.
>
> > Some part of the writing is not clear to me.
>
> We have made revisions in bleue based on your specific comments to improve the clarity of the writing.
>
> > What is deduplication?
>
> Deduplication is a crucial step in ensuring the statistical validity of our contamination detection test. It is detailed in section 3.2 and showcased in Algorithm 1. We added a new reference to section 3.2 when dedup is mentioned for better clarity.
>
> Specifically, our statistical test for radioactivity relies on the assumption that the score for each "scored token" (or in our case, each scored context window) is an independent and identically distributed (i.i.d.) trial. This assumption is vital for the p-value calculation, which assumes that under H_0, the score follows a binomial distribution $S \sim B(\tilde{N}, 1/2)$, where $\tilde{N}$ is the total number of scored trials.
>
> To ensure each trial is independent and identically distributed (i.i.d.), we only score each unique context window once (following Fernandez et al., 2023). This count should follow a binomial distribution, $S \sim B(\tilde{N}, 1/2)$ under H0, and a significantly high score allows us to reject H0 and conclude the model is contaminated. The corresponding p-value is calculated using the regularized incomplete Beta function: p-value(s) = $I_{0.5}(s, \tilde{N} - s + 1)$.
>
> > If the next token is almost deterministically identified based on the previous token, artificially increasing the probabilities of the green tokens as proposed may make the rephrased question invalid. Does this never happen?
>
> The watermark strength in the rephrasing process is a trade-off. $\delta$ is added to the loggits, and top-p is applied at the end. If something is very unlikely, it should remain unlikely (if delta is not too big). This issue can be addressed through low watermark bias ($\delta$). However,  our results (Figure 3a) show that the utility of the rephrased benchmarks is preserved, as the relative ranking of models remains consistent, which is strong indirect evidence that the semantic quality has not been catastrophically damaged. Using a more powerful, instruction-tuned LLM for rephrasing, could also be better at maintaining fluency and coherence even with the biased decoding: an example where a 8B fails and a 70B succeeds is shown in Figure 6 of the appendix.
>
> We acknowledge the possibility of invalid output and recommend that benchmark creators perform quality checks (as discussed in our response to Reviewer Da2v) and tune $\delta$ to find the optimal balance between watermark strength and text quality for their specific domain.

---

> > ### Author Response · Authors · 2025-11-21
> >
> > > Lines 270--271: We train 1B transformer models (Vaswani, 2017) Do the authors use the exactly the same architecture and the training objective as in Vaswani et al. (2017)? (Also, please correct this bibliography item. The other authors are missing.)
> >
> > Thank you for catching the incomplete citation. We corrected the reference to include all authors of "Attention Is All You Need."
> >
> > We use a standard decoder-only transformer architecture with the causal language modeling objective (next-token prediction), which is the modern standard for LLMs, derived from the decoder-only setup proposed in Vaswani et al. (2017). We added this clarification. For the 1B model, we use a hidden dimension of 2048, 25 layers, and 16 attention heads. We clarified this in the paper.
> >
> > > In the experiment of Table 4, why do the authors limit the k so small? I find the proposed k quite small. Why don't you test larger ones?
> >
> > k is the watermark window size.
> >
> > We limited $k$ to values 0, 1, and 2 because $k$ controls the robustness of the watermark. Specifically, $k$ determines the context size over which the green list is generated.
> >
> > A smaller window size $k$ leads to a more robust watermark  (Kirchenbauer et al. 2023). A larger $k$ means the cryptographic link between a token and the green list is based on a longer history. Consequently, an attacker attempting to break the watermark (e.g., via paraphrasing) only needs to alter, on average, $1/k$ of the tokens to break the cryptographic lock for the subsequent tokens. Therefore, using small $k$ values maximizes the robustness of the watermark signal against potential minor alterations or noise during contamination. Our choices of small $k$ values are aligned with maximizing the watermark's inherent robustness, and what is used in pracive (for instance in SynthID text, see [1]).
> >
> > However, if the reviewer thinks that it would be valuable to test for higher values, we could add an additional experiment.
> >
> > > Lines 286--287: What are "the different choices" and why do we want to choose "the one with the smallest loss" for the evaluation?
> >
> >
> > The benchmarks used, ARC and MMLU, are multi-choice question-answering (QA) benchmarks. For these benchmarks, the "different choices" refer to the set of candidate answers provided for each question.
> >
> > The standard method to evaluate a model's performance on these benchmarks is to compute the loss of the correct answer when conditioned on the question and compare it to the losses of the incorrect answers. Specifically:
> >
> > We compute the average negative log-likelihood (loss) of the entire sequence of the proposed answer conditioned on the question for each candidate choice.
> > The model's prediction is the choice that results in the smallest loss, meaning the model assigns the highest probability to that answer being the correct continuation of the prompt.
> >
> > This methodology is standard practice for evaluating LLMs on multiple-choice QA tasks and ensures a fair, log-probability-based assessment of the model's knowledge, which is less noisy than generating the final answer. We follow the convention used in established evaluation frameworks like LM-Harness. We added a clarification in the revised manuscript.
> >
> > > Lines 279--280: Between steps 2500 and 7500, every 5000/\#contaminations, we take a batch from the shuffled concatenation of the three benchmarks instead of the batch from DCLM. This sentence is unclear. I believe it requires rewriting.
> >
> > We agree that the sentence is poorly phrased. We clarified it in blue.
> >
> > Revised Text: “Between training steps 2,500 and 7,500, we perform contamination by replacing a training batch from the DCLM corpus with a batch sampled from the shuffled concatenation of the three watermarked benchmarks. This batch replacement occurs periodically, specifically every 5000/(# Contamination Steps) training steps”.
> >
> > > "Inferential evidence": I think I understand the nuance, but I would like to suggest avoiding the word "inferential" in such an informal sense, especially in research on artificial intelligence. Do the authors believe these words are appropriate here?
> >
> > We appreciate this note. We used "inferential" to mean that existing methods are empirical and are unable to provide proof of contamination, contrasting it with the $p$-value we provide, and used “inferential” as it is used in the fields of privacy and security. However, we agree it risks confusion. We can revise the text to use more precise language, such as "heuristic evidence," , “empirical evidence”, "circumstantial evidence," or "post-hoc analysis," to draw the necessary contrast with our provable, statistical guarantee. Does that seem good?
> >
> > > Table 1 (and other similar experiments): What is "the number of condemnations"? (What is the unit?)
> >
> > We added clarification in the manuscript (section 4.2, contamination set-up). Specifically, number of contaminations is the number of times that benchmark was seen during training, with the dection set-up described in this section.

---

> > > ### Author Response · Authors · 2025-11-21
> > >
> > > > Lines 285--305: what is the difference between the setups in calculating the p-values in Table 1 (and Table 6) and Figure 3b (and Figure 8)? I suppose the radioactivity score $S$ was only used in the Contamination detection experiment.
> > >
> > > The setups are fundamentally the same, as the $p$-value is calculated directly from the radioactivity score $S$. The radioactivity score $S$ is the test statistic in the contamination detection experiment. The underlying calculation is identical: the $p$-value is derived from the radioactivity score $S$.
> > >
> > > We can answer additional questions if that is not clear.
> > >
> > > > Lines 350--352: Table 1 links the contamination detection to the actual cheating on the benchmarks when $\delta=4$ is used. We see that for the three benchmarks, whenever the cheat is greater than 10%, detection is extremely confident. How do I see this from Table 1?
> > >
> > > In Table 1, for each benchmark column, the ‘Acc.’ sub-column shows the performance of the model at the end of training, for different numbers of contaminations. In light grey, we see how many percentage points the model has gained compared to the non-contaminated run. We can see that when the gain is around 10%, for all benchmarks, the corresponding p-value of the detection test is very small, and that we can therefore flag contamination with high confidence (low FPR). We added this clarification in the revised version.
> > >
> > > Note: A continuous version of this experiment can be observed in Figure 4.
> > >
> > > > "Figure 4.4": Figure 4?
> > >
> > > Yes, this is a typo. It should be "Figure 4." Thank you for catching it, we corrected it.
> > >
> > > > Line 411--412: "larger models require fewer contaminated batches to achieve the same gain on the benchmark". Is this something we can see from Figure 4?
> > >
> > > The “number of contaminated batches” is the "number of contaminations" as explained before to the reviewer. It refers to how many times the model was exposed to the benchmark during pretraining. Figure 4 demonstrates that contamination leads to a $+6%$ gain on the benchmark across different set-ups: $4$ contaminations for the $1$B model, $8$ for the 360M model, and $16$ for the $135$M model. This finding indicates that the $1$B model requires four times fewer exposures than the $135$M model to achieve the same performance boost. Crucially, detection tests consistently yield p-values around $10^{-5}$ for all three setups. This consistency highlights that the detection method accurately measures the artificial gain on the benchmark, rather than simply measuring the number of contaminations.
> > >
> > > We clarified this by changing “number of contaminated batches” to “number of conatminations” to be more consistent.
> > >
> > > > In Figure 5: Question: "The rate of acceleration of an object is determined by the mass of the object and" Is this question complete?
> > >
> > > Yes, that is the complete (truncated) prompt used for the experiment. It is a common format in multiple-choice questions where the model is expected to generate the rest of the sentence or the correct answer/option based on this incomplete context. The specific prompt is from the ARC dataset, where the model must choose the correct option.
> > >
> > > > In Table 4: "dedup" --> deduplication?
> > >
> > > Yes, "dedup" is an abbreviation for "deduplication." We corrected this.
> > >
> > > We thank the reviewer for their time in evaluating our work, and remain available for further discussion
> > >
> > > [1] Dathathri, Sumanth, et al. "Scalable watermarking for identifying large language model outputs." Nature 634.8035 (2024): 818-823.

---

> > > > ### Comment · Reviewer_EUzr · 2025-11-27
> > > >
> > > > I thank the authors for their responses to my comments. I have a few follow-up comments.
> > > >
> > > > > we can clarify this by explicitly stating that our method is designed for models that operate on a next-token prediction objective
> > > >
> > > > Yes, I encourage the authors to do so. Whether next-token prediction is the current SOTA paradigm does not change the fact that the method may not extend beyond that framework. I am not suggesting this is a critical flaw of the method itself, but the paper is weakened by the lack of clarity about its scope.
> > > >
> > > > > the decoder-only setup proposed in Vaswani et al. (2017)
> > > >
> > > > As far as I know, Vaswani et al. (2017) did not propose a decoder-only setup. I request that the authors ensure the correctness of the citations.
> > > >
> > > > > We can revise the text to use more precise language, such as "heuristic evidence," , “empirical evidence”, "circumstantial evidence," or "post-hoc analysis," to draw the necessary contrast with our provable, statistical guarantee. Does that seem good?
> > > >
> > > > Yes, any of the suggested changes sounds fine to me.
> > > >
> > > > > A smaller window size leads to a more robust watermark
> > > >
> > > > I find it a little confusing because the authors response seems to advocate for a smaller window size, while Table 4 shows that the p-value is smaller (thus better) for a larger window.
> > > > Also, if smaller is better, why not take the smallest? Maybe there is a trade-off, but the experiment fails to show that because the range of k is so small. Figure 4 rather seems to suggest that larger is better.

---

> ### Author Response · Authors · 2025-11-28
>
> Dear reviewer,
>
> Thank you for your answer.
>
> > Yes, I encourage the authors to do so. Whether next-token prediction is the current SOTA paradigm does not change the fact that the method may not extend beyond that framework. I am not suggesting this is a critical flaw of the method itself, but the paper is weakened by the lack of clarity about its scope.
>
> We added this in blue line 147 in the "scope" paragraph.
>
> > As far as I know, Vaswani et al. (2017) did not propose a decoder-only setup. I request that the authors ensure the correctness of the citations.
>
> You are absolutely correct. We clarified this by citing T-Decoder and GPT in line 287: ``Training set-up. We train 1B standard decoder-only transformer architecture with the causal language modeling objective similar to the foundational T-decoder (Liu et al., 2018) and GPT (Radford et al., 2018), adapted from the encoder-decoder architecture proposed in Vaswani et al. (2017) using Meta Lingua (Videau et al., 2024) and the code-base’s default architectural choices. We train on 10B tokens from DCLM (Li et al., 2024), with the same tokenizer used to embed the watermark, unless stated otherwise (e.g., in Sec. 4.5). The model architecture includes a hidden dimension of 2048, 25 layers, and 16 attention heads. "
>
> We hope these clarifications are satisfying; we agree that correctly citing is very important!
>
> > Yes, any of the suggested changes sounds fine to me.
>
> We replaced by heuristics evidence instead of inferential.
>
> > I find it a little confusing because the authors response seems to advocate for a smaller window size, while Table 4 shows that the p-value is smaller (thus better) for a larger window. Also, if smaller is better, why not take the smallest? Maybe there is a trade-off, but the experiment fails to show that because the range of k is so small. Figure 4 rather seems to suggest that larger is better.
>
> Smaller window sizes lead to more robust watermarks, but it leads to more visible biases (increasing the frequency of tokens instead of k=0 compared to 2-grams when k=1 etc). In our experiments, because of deduplication, the test is more powerful with k=2 than k=0 or k=1 as it allows to score more tokens. So we choose k=2 as a good tradeoff: it enhances contamination detectability while still being in a good spot for watermark robustness.
>
> We thank again the reviewer for their time and remain available for further discussion

---

### Official Review · Reviewer_R8oW · 2025-10-30

**Soundness:** 3
**Presentation:** 3
**Contribution:** 3
**Rating:** 6
**Confidence:** 4

**Summary:**

This paper proposes a proactive framework to detect benchmark contamination in LLMs. Instead of trying to find evidence of contamination after a model is trained, the authors suggest embedding a cryptographic watermark into a benchmark before its public release. This is achieved by using an LLM to rephrase the benchmark's questions, embedding a secret statistical signal (the watermark) during the text generation process.
To test a suspect model, a benchmark provider can then check for the presence of this watermark's statistical bias in the model's predictions. The method requires white-box (logit) access to the suspect model and includes a novel algorithm to handle mismatches between the tokenizers used for watermarking and by the suspect model. Through extensive experiments involving pre-training 1B-parameter models with controlled contamination on benchmarks like MMLU and ARC, the authors demonstrate that their method preserves the benchmark's utility while detecting contamination with very high statistical confidence.

**Strengths:**

- Proactive and Verifiable: The core strength is its shift from post-hoc, inferential detection methods to a proactive approach that provides verifiable, statistical proof (a p-value) of contamination. Meanwhile, the experiments show that even with a strong watermark, the rephrased benchmarks remain effective for evaluating and ranking models, with performance being very similar to the original versions.
- Practicality: The paper addresses the practical challenge of distinct tokenizers by introducing a robust cross-tokenizer detection algorithm, making the solution more broadly applicable.

**Weaknesses:**

- White-Box Access Requirement: The detection test requires full logit access to the suspect model. This limits its use to open-source models and cannot be used by external parties to audit closed, API-only models.
- Vulnerability to Intentional Evasion: The framework is primarily designed to detect unintentional contamination from sources like web scraping. A determined, malicious actor could potentially devise strategies to circumvent detection, such as by rephrasing the questions again to remove the watermark or by training only on the answers.

**Questions:**

- The experiments focus on question-answering benchmarks. How do you envision this watermarking approach being applied to other types of evaluation, such as code generation, mathematical reasoning (where rephrasing is brittle), or creative writing tasks?
- The method focuses on watermarking the questions. Have you considered the possibility of watermarking the provided answers for open-ended questions?

---

> ### Author Response · Authors · 2025-11-21
>
> We are very grateful for the reviewer's positive assessment of our paper's proactive and verifiable framework. We address the noted weaknesses and questions.
>
> > White-Box Access Requirement: The detection test requires full logit access to the suspect model. This limits its use to open-source models and cannot be used by external parties to audit closed, API-only models.
>
> We believe that the white-box is widely applicable and relevant for two primary reasons:
>
> - Open-Source Models: Our method is directly applicable to the ecosystem of powerful open-source models.
> - Internal Auditing: It is a vital tool for closed-source model owners for internal audits to guarantee the integrity of their reported benchmark scores. Compared to other approaches, we are the only approach that provides p-values without the necessity of needing private held-out sets: anyone can run the contamination detection test on their hands.
>
> While black-box detection (e.g., detecting the watermark in the model's completion of a watermarked prompt, as explored in prior work) is possible, it is significantly less effective and would not provide the strong statistical guarantees offered by our white-box approach.
>
> > Vulnerability to Intentional Evasion: The framework is primarily designed to detect unintentional contamination from sources like web scraping. A determined, malicious actor could potentially devise strategies to circumvent detection, such as by rephrasing the questions again to remove the watermark or by training only on the answers.
>
> Indeed, defending against a determined, malicious actor is a challenging problem. We state in the manuscript as as one of the main limitations Our primary goal is to provide a robust defense against unintentional, large-scale pre-training contamination resulting from web scraping.
>
> However, our framework provides inherent resistance to the evasion strategies mentioned as is uses provably robust existing watermarks. Indeed, as we demonstrate in our ablations, the strength of detection is directly proportional to the proportion of green tokens in the watermarked text. An adversarial rephrasing attack inherently alters the token sequence, which weakens the watermark signal (reducing the green token proportion). A successful attack would result in a high (non-significant) $p$-value, correctly reflecting that the contamination signal has been diluted or destroyed.
>
> About training only on answers: to mitigate this, the watermarking strategy could be expanded to include the watermarking of the answers if possible or a context surrounding the answer (e.g., adding a watermarked instruction or prompt template) to ensure that the watermarked tokens are present in the training sequence used to condition the model's output.
>
> > The experiments focus on question-answering benchmarks. How do you envision this watermarking approach being applied to other types of evaluation, such as code generation, mathematical reasoning (where rephrasing is brittle), or creative writing tasks?
>
> The core principle is universally applicable across textual datasets, provided a watermark can be embedded into the training data.
>
> Code/Math/Summarization: Rephrasing the question prompt might be too brittle for tasks where input precision is key (like math), but still possible (see [A]). In these cases, we could do watermarking of the surrounding context. For example, a watermarked instruction template (e.g., "Solve the following problem, showing your steps. [WATERMARKED TEXT]") or a watermarked header/footer can be added to the input prompt, which would be included in the pre-training corpus. The model would still be evaluated on the output, but the presence of the watermarked context in its training data would be detectable.
>
>
> Creative Writing/Dialog: The method can be applied by watermarking the initial prompt/stimulus or dialogue turns that the model is expected to learn from. The statistical test's power depends only on the total number of watermarked tokens the model has memorized, making it a general solution for any textual data.
>
> > The method focuses on watermarking the questions. Have you considered the possibility of watermarking the provided answers for open-ended questions?
>
> Yes. For benchmarks with free-form or open-ended answers (e.g., summarization, code generation, creative writing), watermarking the provided reference answers is also a viable strategy.
>
> If a model is trained on the full prompt-answer pair, it will memorize the watermark present in the reference answer just as effectively as the watermark in the question. This expands applicability, especially for tasks where rephrasing the question is difficult.
>
> This is discussed this in section 3.1.
>
> We thank the reviewer for their time and remain available
>
> [A] Lee, Taehyun, et al. "Who wrote this code? watermarking for code generation." Proceedings of the 62nd Annual Meeting of the Association for Computational Linguistics (Volume 1: Long Papers). 2024.

---

> > ### Comment · Reviewer_R8oW · 2025-11-28
> >
> > Thanks for your response. I would like to keep my score and remain positive towards acceptance.

---

### Official Review · Reviewer_uLh1 · 2025-11-01

**Soundness:** 3
**Presentation:** 3
**Contribution:** 3
**Rating:** 4
**Confidence:** 4

**Summary:**

The paper introduces a method that offers provable evidence of benchmarks for LLMs. The key idea involves watermarking benchmarks before public release, achieved through rephrasing LLMs. Contamination in downstream models is then reliably identified using a hypothesis test that leverages the "radioactivity" inherent in watermarks. The authors successfully validate this approach, demonstrating its efficacy by detecting contamination in an LLM pre-trained on a contaminated corpus. Furthermore, they confirm that the rephrasing process maintains the original utility of the benchmarks.

**Strengths:**

* The paper is well-written, easy to follow, and features comprehensive, well-executed experiments.
* The proposed technique is demonstrably effective. To confirm its reliability and understand how the repetition factor influences the strength of the statistical test, the authors trained multiple LLMs from scratch on a pre-contaminated corpus.
* The authors propose a novel detection algorithm to detect radioactivity in case of tokenizer mismatch between the rephrasing LLMs and the suspect LLMs. This enhances the practical applicability of the algorithm.

**Weaknesses:**

* The method overlaps significantly with past works that were not used as baselines or mentioned in related works [1,2,3,4]. For example, [1] also proposes a similar idea and proposes a hypothesis test that leverages watermarked rephrases. [2] proposes a dataset inference method that can also be used for detecting contamination. Works such as these should be compared in related works and evaluated against as benchmarks.


* The current demonstration, while confirming the utility of the rephrased benchmarks across various LLM sizes, is limited because all tested LLMs belong to the same model family as the one used for the original rephrasing. It is essential that the authors extend their evaluation to include LLMs from different model families.


* For reliable detection at a statistically significant strength (requiring approximately 16 repetitions as shown in Figure 3b), a substantial amount of repetition is necessary. Furthermore, it would be beneficial to include a demonstration of how the size of the total training corpus influences the test's strength. The detection also requires benchmarks of significant sample size (around 2k-4k) samples.


* The authors should also empirically verify that their method is robust against false positives using off the shelf LLMs. For a more rigorous test of false positives, the detection strength should also be evaluated against keys that are distinct from the contaminated keys.


I would be willing to reconsider my overall assessment based on authors' responses to the above points. For instance, based on favorable comparisons with highly relevant prior work.

References:

[1] STAMP Your Content: Proving Dataset Membership via Watermarked Rephrasings. ICML 25

[2] LLM Dataset Inference: Did you train on my dataset?

[3] DE-COP: Detecting Copyrighted Content in Language Models Training Data

[4] Proving Test Set Contamination in Black Box Language Models

**Questions:**

* Will the proposed method be applicable to other general forms of dataset?

* Could the authors include the % percentage of scored tokens in Table 5?

* The method does not seem to be applicable on existing benchmarks, can I suggest that the paper acknowledges this limitation in their work?

---

> ### Author Response · Authors · 2025-11-21
>
> We thank the reviewer for their positive assessment of our paper's soundness, presentation, and contribution, and for providing constructive feedback. We address the noted weaknesses and questions below.
>
> In summary, we
> - added related works
> - ran STAMP detection and found issues with FPR
> - Show that other families of models also have the same performance on all watermarked benchmarks
> - Ran experiments on the impact of training data-set size (we will add the results once available)
> - Added an experiments for FPR by showing that p-values are correctly uniformly distributed when testing with other keys, as asked by the reviewer
>
> > The method overlaps significantly with past works that were not used as baselines or mentioned in related works \[1,2,3,4\]. For example, \[1\] also proposes a similar idea and proposes a hypothesis test that leverages watermarked rephrases. \[2\] proposes a dataset inference method that can also be used for detecting contamination. Works such as these should be compared in related works and evaluated against as benchmarks.
>
> STAMP [1] is concurrent work and was published at the same time at ICML 2025 that 2 months before the ICLR submission. From official ICLR guidelines, we should not compare them.
> Moreover, we would like to emphasize that STAMP differs significantly from our work as it requires keeping private versions of the benchmarks. Our method only requires one public version, which is our main motivation, as stated in the abstract and introduction. This means that, similar to other Membership Inference Attack (MIA) methods, contamination detection for STAMP can only be performed by the person who owns the private versions of the benchmarks. Since it necessitates open-weight access to the model (or at least grey-box access with logits), in practice, model owners cannot use this test for detection without disclosing their models. We argue that this is not a fair comparison as the set-up is very different from ours.
>
> Still, we tried to compare, following the reviewer’s request. We replicated our setup: rephrased MMLU* (5k questions from MMLU) using Llama-3.1-8B, with $\delta = 4$. We followed STAMP’s methodology and constructed 5 different versions with 5 different secret keys (the number 5 is used in the authors' main experiments). Running STAMP, we obtained a p-value of $7.9 \times 10^{-12}$ on a non-contaminated model, which should happen with a probability of $1 / 100\text{B}$. Although the paper has a section on False Positive Rate (FPR), it does not specify the exact setup (Is 5 private versions enough to get a reliable test? Are the presented results averaged across keys?). We have contacted the authors to clarify the replication details. At present, obtaining such a low p-value under $H_0$ makes it impossible for us to perform a meaningful comparison under $H_1$, as the $p$-value is nonsensical (we do get even lower p-values after contamination, but it cannot be interpreted as p-values as they are too low under H0!). We have added STAMP to the related work section. We also added [2] to the related work section,  but it again necessitates having data from the same distribution as the benchmarks, which constitutes a very different setup from ours. We also now mention 3 and 4. We thank the reviewer for pointing out the references.
>
>
> We have updated the related work section, and also the “comparaison to other methods” subsection 4.2.
>
> > The current demonstration, while confirming the utility of the rephrased benchmarks across various LLM sizes, is limited because all tested LLMs belong to the same model family as the one used for the original rephrasing. It is essential that the authors extend their evaluation to include LLMs from different model families.
>
> We agree that evaluating across different model families strengthens the generality of our findings. We conduct an additional experiment with the Gemma and Qwen families, added a figure in the appendix. We observe that it keeps the performance for all models. This is a table:
>
> | Model Family | Model Name | delta = 0 |  delta = 1 |  delta = 2 |  delta = 4 |
> | :--- | :--- | :---: | :---: | :---: | :---: |
> | **Google Gemma 3** | `gemma-3-1b-pt` | 68.01% | 68.10% | 68.43% | 68.98% |
> | | `gemma-3-4b-pt` | 76.68% | 76.47% | 76.47% | 76.26% |
> | | `gemma-3-12b-pt` | 81.99% | 82.37% | 82.37% | 82.53% |
> | | `gemma-3-27b-pt` | 82.28% | 82.32% | 82.28% | 82.45% |
> | **Meta Llama** | `Llama-3.2-1B` | 62.37% | 62.37% | 62.25% | 62.46% |
> | | `Llama-3.2-3B` | 69.28% | 69.49% | 69.57% | 69.74% |
> | | `Llama-3.1-8B` | 76.18% | 75.72% | 75.59% | 75.46% |
> | **Qwen 2.5** | `Qwen2.5-7B` | 74.07% | 73.65% | 73.57% | 74.03% |
> | | `Qwen2.5-14B` | 75.84% | 75.76% | 75.93% | 76.05% |
> | | `Qwen2.5-32B` | 73.78% | 73.99% | 74.07% | 74.12% |

---

> ### Author Response · Authors · 2025-11-21
>
> > For reliable detection at a statistically significant strength (requiring approximately 16 repetitions as shown in Figure 3b), a substantial amount of repetition is necessary. Furthermore, it would be beneficial to include a demonstration of how the size of the total training corpus influences the test's strength. The detection also requires benchmarks of significant sample size (around 2k-4k) samples.
>
> We agree that the size of the total training corpus is a critical factor. Our current contamination experiments were conducted on a fixed 10B token corpus to ensure controlled comparisons. We expand our analysis to show the inverse relationship between corpus size and detection confidence:
>
> Specifically, we train on 5B, 10B, 15B or 20B tokens. We contaminate between 25% and 75% of the training run, 16 contaminations at a regular frequency. We will add a table once available.
>
> [Edit: the results might not be ready before the rebuttal deadline because of cluster issues. We commit to include them in the camera-ready version if accepted]
>
>
> > The authors should also empirically verify that their method is robust against false positives using off the shelf LLMs. For a more rigorous test of false positives, the detection strength should also be evaluated against keys that are distinct from the contaminated keys.
>
> This is a crucial point for demonstrating the method's reliability. Our current false-positive analysis is implicit in the p-values reported for non-contaminated models (e.g., the base model, 0-reps in Figure 4), and show that it gives random p-values as expected under H0
> We add an experiment with 294 other secret keys, and show that the p-values are uniformly distributedL
>
>
> | Metric | Value |
> | :--- | :--- |
> | **Descriptive Statistics** | |
> | Number secret-keys tested ($N$) | 294 |
> | Mean | 0.5113 |
> | Standard Deviation | 0.3116 |
> | **Kolmogorov-Smirnov Test for Uniformity** | |
> | KS Statistic | 0.0614 |
> | P-value | 0.2094 |
> | **Conclusion** | **Fail to reject $H_0$* (Distribution appears uniform) |
>
>
> > Will the proposed method be applicable to other general forms of dataset?
>
> Yes, the method is broadly applicable. The core requirement is that the dataset text can be modified to embed a watermark, either by rephrasing the text itself (as done with the QA questions) or by adding watermarked context around the data. The statistical test's power depends on the total number of watermarked tokens the model has memorized, making it a general solution for any textual dataset used in pre-training. We added this in the conclusion.
>
> > Could the authors include the % percentage of scored tokens in Table 5?
>
> We add the percentage of scored tokens (i.e., the tokens used to compute the radioactivity $p$-value) in the revised Table 5. The number of tokens in the questions of the corresponding watermarked version of MMLU* is 332k (with the tokenizer for llama3). The number of tokens scored is because of the filtering due to a potential mismatch in tokenizer in the suspect model, as well as the deduplication phase
>
> > The method does not seem to be applicable on existing benchmarks, can I suggest that the paper acknowledges this limitation in their work?
>
> We agree that a key aspect of our approach is its proactive nature, meaning it is effective when the benchmark is watermarked before its first public release. We explicitly state this in the abstract and in the method section, but added this point to the limitations
>
>
> We thank again the reviewer for their time and remain available for additional discussion.

---

### Meta-Review · Area_Chair_xiMu · 2026-01-06

**Summary:**

The paper addresses the important problem of benchmark contamination and proposes a proactive watermarking-based detection framework with solid empirical validation.

However, multiple reviewers raised concerns that the core idea is a straightforward application of existing watermarking “radioactivity” and keyed rephrasing techniques to a slightly different setting, with limited conceptual novelty.

While the experiments are careful, the contribution does not sufficiently advance the underlying methodology to meet the ICLR novelty bar.

**Reviewer Concerns:**

The authors addressed several technical and presentation issues in the rebuttal, including additional experiments across model families, false-positive analysis, and clearer scope and limitations. Nonetheless, the main concern regarding incremental novelty relative to prior work remains unresolved. The paper largely repackages established watermarking ideas for benchmark governance without introducing a fundamentally new mechanism or insight.

**Reviewer Scores:**

Reviewers uLh1 and gZm1 would likely remain near their original borderline-to-negative scores, as their novelty-related concerns were only partially alleviated. Reviewers EUzr and R8oW maintained positive scores but explicitly indicated openness to rejection. Overall, the score distribution remains mixed and does not converge to a confident acceptance.

---

### Decision · Program_Chairs · 2026-01-26

Reject